# Magnetic Entropic Forces Emerging in the System of Elementary Magnets Exposed to the Magnetic Field

**DOI:** 10.3390/e24020299

**Published:** 2022-02-20

**Authors:** Edward Bormashenko

**Affiliations:** Chemical Engineering Department, Engineering Sciences Faculty, Ariel University, Ariel 407000, Israel; edward@ariel.ac.il

**Keywords:** entropic force, magnetic field, linear system of elementary magnets, ordering, temperature, repulsion force

## Abstract

A temperature dependent entropic force acting between the straight direct current *I* and the linear system (string with length of *L*) of *N* elementary non-interacting magnets/spins $\vec{\mu }$ is reported. The system of elementary magnets is supposed to be in the thermal equilibrium with the infinite thermal bath *T*. The entropic force at large distance from the current scales as $F_{magn}^{en}\sim \frac{1}{r^{3}}$, where *r* is the distance between the edge of the string and the current *I*, and $k_{B}$ is the Boltzmann constant; (r≫L is adopted). The entropic magnetic force is the repulsion force. The entropic magnetic force scales as $F_{magn}^{en}\sim \frac{1}{T}$, which is unusual for entropic forces. The effect of “entropic pressure” is predicted for the situation when the source of the magnetic field is embedded into the continuous media, comprising elementary magnets/spins. Interrelation between bulk and entropy magnetic forces is analyzed. Entropy forces acting on the 1D string of elementary magnets that exposed the magnetic field produced by the magnetic dipole are addressed.

## 1. Introduction

So-called entropic forces has attracted the attention of investigators in last few decades. An entropic force acting in a system is an emergent phenomenon resulting from the entire system’s statistical tendency to increase its entropy [1,2]. Entropic force represents the tendency of a system to evolve into a more probable state, rather than simply into one of lower potential energy [1,2]. A classic example of an entropic force is the temperature dependent elasticity of a freely-jointed polymer chain [3,4]. For an ideal polymer chain, maximizing its entropy means reducing the distance between its two free ends [3,4]. Consequently, an entropic elastic force that tends to collapse the chain is exerted by the ideal chain between its two free ends [3,4]. Muscles of mammals are also driven by entropy forces [5]. As it has been shown, elasticity in the giant muscle protein titin arises from entropy in a way very similar to the entropy-driven elasticity of polymer chains [5]. The so-called “hydrophobic effect” represents additional exemplification of the entropy-forces-driven phenomena. The hydrophobic interaction originates from the disruption of hydrogen bonds between molecules of liquid water by the nonpolar solute [6]. By aggregating together, nonpolar molecules reduce the surface area exposed to water and minimize the effect [6]. This reducing of the surface is thermodinamically (entropically) favorable, giving rise to the clustering of small hydrophobic particles [6]. An interest to entropic force was strengthened after the suggestion of Verlinde, who hypothesized the entropic nature of gravity [7]. The entropic origin of gravity was discussed in detail in Refs. [8,9]. It was shown that classical Newtonian gravity may be interpreted in terms of an entropic force [8,9]. The entropy origin of gravity was criticized in Refs. [10,11,12], and the problem remains open and debatable. Motivated by Verlinde’s theory of entropic gravity, a tentative explanation to the Coulomb’s law with an entropic force was suggested [13]. We demonstrate the temperature dependent magnetic entropic forces emerging when a string of elementary magnets is exerted to the magnetic field, which tends to order the magnets and in turn to diminish the entropy of the system. 

## 2. Results and Discussion

### 2.1. Thermodynamics of Magnetics: Origin of Entropy Forces

The general expression for the Helmholtz free energy Φ of a magnetic material exposed to the external magnetic field is supplied by Equation (1): (1)$$d\Phi =-SdT-TdS+\zeta dN+\frac{1}{2}\vec{H}\cdot \vec{B}dV\;$$
where *S*, *T*, and *V* are the entropy, temperature, and volume of the magnetic body correspondingly, $\vec{H}$ and $\vec{B}$ are the magnetic field and magnetic flux intensities, and ζ and *N* are the chemical potential and the number of particles in the magnetic body correspondingly [14,15]. Consider the isothermal magnetic body (T=const) when the number of the particles *N* is fixed. In this case, we obtain: (2)$$d\Phi =-TdS+\frac{1}{2}\vec{H}\cdot \vec{B}dV\;$$

The first term in Equation (2) will give rise to the so-called entropic forces [4–6}, whereas the second term is responsible for the magnetic forces which we label in our text “the bulk magnetic forces”, or for purposes of brevity “bulk forces”. The bulk forces are addressed in detail in Ref. [14]. Consider that, in the classical textbook by Landau and Lifshitz [14], the thermo-isolated magnetic is analyzed in detail and consequently dS=0 is implied. We treat in our paper the “entropic term” of Equation (2) and the entropic forces emerging from this term, under comparison of the entropic magnetic forces to the bulk ones. 

### 2.2. Entropic Magnetic Forces Acting on a Magnetic Body Appearing in the Field Produced by Infinite Direct Current

Consider the linear (1D) string of elementary non-interacting magnets (spins) $\vec{\mu }$ exerted to the magnetic field generated by a straight, infinite, direct current located, as depicted in Figure 1. 

We assume that there are *N* separate and distinct sites fixed in a space and aligned, as shown in Figure 1. Attached to each site is an elementary magnet, which can point only up or down, as shown in Figure 1. The total length of the string is *L*, and the linear density of the magnets $\tilde{N}$, defined according to Equation (3), is supposed to be constant along the string:(3)$$\tilde{N}=\frac{N}{L}=const$$

The suggested 1D string built of elementary magnets/spins $\vec{\mu }$ is embedded into magnetic field $\vec{H}$ generated by the infinite straight current $H\left(r\right)=\frac{I}{2\pi r}$, as shown in Figure 1, leading to the spin orientation. The potential energy of a single elementary magnet in the magnetic field is given in the SI system of units by Equation (4):(4)$$U\left(r\right)=-\mu_{0}\vec{\mu }\cdot \vec{H}\left(r\right)$$
where $\mu_{0}$ is the vacuum permeability. Assume also that the system of spins is in the thermal equilibrium with the surrounding (thermal bath) under the constant temperature *T* (the system is isothermal). Let us divide the string of the magnets into “sub-strings” as follows: let $dN_{i}$ be the number of spins in the sub-string $dr_{i}$ numbered “*i*”, the magnetic field within the string is $H\left(r_{i}\right)=\frac{I}{2\pi r_{i}}$ (distance $r_{i}$ is shown in Figure 1). The entropy of the sub-string $S_{i}$ was addressed in detail in [16,17], and within the approximation of the weak magnetic field, i.e., when $\mu_{0}\mu H\ll k_{B}T$ takes place ($k_{B}$ is the Boltzmann constant and *T* is the temperature), the entropy is given by Equation (5) (consider $dN_{i}=\tilde{N}dr_{i})$: (5)$$S_{i}=S_{0i}-\frac{\mu_{0}^{2}\mu^{2}H^{2}r_{i}dN_{i}}{2k_{B}T^{2}}=S_{0i}-\frac{\tilde{N}\mu_{0}^{2}\mu^{2}I^{2}dr_{i}}{8\pi^{2}k_{B}T^{2}r_{i}^{2}}$$
where $dr_{i}$ is the length of the “*i*th” sub-string, $dN_{i}$ is the number of spins in the “*i*th” sub-string $dr_{i}$, and $S_{0i}=k_{B}ln\frac{N_{i}!}{\left(\frac{1}{2}N_{i}!\right)\left(\left(\frac{1}{2}N_{i}!\right)\right)}≅k_{B}\left[N_{i}ln2-\frac{1}{2}ln\frac{2}{\pi N_{i}}\right]$ (in the Stirling approximation) is the constant. It is latently adopted that a sub-string contains a “large” number of spins, enabling the statistical approach; for details, see [16,17], in which the field of validity of Equation (5) is carefully addressed. The total entropy S of the string of spins exerted to the magnetic field generated by the infinite straight direct current I, depicted in Figure 1, is given by Equation (6): (6)$$S=\overset{n}{\underset{i=1}{\sum }}S_{i}=S_{0}-\int_{r}^{r+L}\frac{\tilde{N}\mu_{0}^{2}\mu^{2}I^{2}}{8\pi^{2}k_{B}T^{2}r^{2}}dr$$
where *n* is the total number of the sub-strings, *L* is the total length of the string, and $S_{0}=\overset{n}{\underset{i=1}{\sum }}S_{oi}$. Integration in Equation (6) yields Equation (7) (consider Equation (3) and $\tilde{N}=\mathrm{cons}t$): (7)$$S=S_{0}-\frac{\tilde{N}L\mu_{0}^{2}\mu^{2}I^{2}}{8\pi k_{B}T^{2}r\left(r+L\right)}=S_{0}-\frac{N\mu_{0}^{2}\mu^{2}I^{2}}{8\pi^{2}k_{B}T^{2}r\left(r+L\right)}$$

The entropic magnetic force $F_{magn}^{en}$ emerging from Equations (2) and (7) is given by Equation (8) (see [3,4]):(8)$$F_{magn}^{en}=-\frac{d\Phi }{dr}=T\frac{\partial S}{\partial r}$$ Substitution of Equation (7) into Equation (8) supplies the expression for the entropic force: (9)$$F_{magn}^{en}=\frac{N\mu_{0}^{2}\mu^{2}I^{2}}{8\pi^{2}k_{B}T}\frac{2r+L}{r^{2}{\left(r+L\right)}^{2}}$$ Equation (7) for the large distances, i.e., r≫L is transformed into the following expression:(10)$$S=S_{0}-\frac{\tilde{N}L\mu_{0}^{2}\mu^{2}I^{2}}{8\pi^{2}k_{B}T^{2}r^{2}}=S_{0}-\frac{N\mu_{0}^{2}\mu^{2}I^{2}}{8\pi^{2}k_{B}T^{2}r^{2}}$$

The entropic magnetic force $F_{magn}^{en}$ for the large distances, i.e., r≫L is given eventually by Equation (11) (see Equation (8)): (11)$$F_{magn}^{en}=\frac{N\mu_{0}^{2}\mu^{2}I^{2}}{4\pi^{2}k_{B}Tr^{3}}$$ The eventual Equation (11) deserves the discussion. First of all, it should be emphasized that the derived magnetic force $F_{magn}^{en}=\frac{N\mu_{0}^{2}\mu^{2}I^{2}}{4\pi^{2}k_{B}Tr^{3}}$ is always the repulsion force; this is intuitively a well-expected prediction. Indeed, whatever is the location of string, the magnetic field always decreases the entropy of the entire string, under ordering of the elementary magnets, which is recognized from Equations (5)–(7), which is thermodynamically unfavorable under isothermal conditions. In addition, the repulsion nature of the entropy magnetic force is independent on the direction of the current *I*, as it is immediately seen from Equations (9) and (11). This result supplies an important prediction: consider the direct current embedded into the continuous medium built of the elementary magnets $\vec{\mu }$, as shown in Figure 2. The magnetic entropic force will repulse elementary magnets, wherever they are located. Thus, in the continuous medium comprising elementary magnets/spins, the phenomenon of the “entropic pressure” stimulated by the external source of the magnetic field (current) is predicted. Recall that the isothermal pressure of an ideal gas is also a pure entropic phenomenon. The entropic force supplied by Equations (9) and (11) will appear for both diamagnetic and paramagnetic materials, seen as ensembles of elementary magnets. 

In the exotic case of negative absolute temperatures, corresponding to the population-inverted regime, when a spectrum of the system is bounded, the magnetic entropic force becomes the attraction force [18,19]; however, the separate discussion of this perplexed situation is demanded [20,21]. 

Secondarily, the entropic magnetic force scales as $F_{magn}^{en}=\frac{const}{T}$, and this is quite surprising for the entropic forces, which are usually growing with temperature [3,4,7,8,9,10,11,12,13]. The entropic elastic force inherent for an ideal polymer chain appears as:(12)$${\vec{F}}_{polym}^{en}=\frac{3k_{B}T}{N_{k}b^{2}}\vec{R}$$
where $N_{k}$ is the number of the Kuhn segments in the chain, *b* is the length of the Kuhn segment, and $\vec{R}$ is the end-to end-distance of the chain [4]. It is seen that this force scales as $F_{polym}^{en}\sim T$, and it is opposite to the temperature scaling law recognized from Equations (9) and (11). However, in our case, this result is quite predictable; consider that the temperature movement withstands the magnetic ordering of spins, and thus prevents their magnetic ordering imposed by the external magnetic field; hence, the temperature-related influence is expected to diminish the magnetic entropic force. It is also seen from Equations (9) and (11) that the entropic elasticity in polymers scales as $F_{polym}^{en}\sim \frac{1}{N_{k}}$, whereas the magnetic entropic force scales as $F_{magn}^{en}\sim N$, which is also indeed quite intuitively clear, and the growth of the total number of the spins strengthens the entire entropic magnetic effect.

Last but not least, the magnetic entropy force scales as $F_{magn}^{en}=\frac{const}{r^{3}}$, and it grows rapidly with the decrease of the distance *r*. The qualitative reasoning for this result is also quite clear: the smaller the distance between the source of the magnetic field and the string of the spins, the larger is the ordering effect; consequently, the larger is decrease in the total entropy of the string, which is thermodynamically unfavorable. 

It also should be taken into account that the magnetic entropic force stems from the entropy gradient (see Equation (8)); the effect of ordering itself will not give rise to the discussed entropic effect. Consider a string of elementary magnets embedded into the ideal solenoid; assume that the string coincides with the axis of the solenoid. The magnetic field produced by the solenoid will order the elementary magnets and an entropy will be uniformly decreased within a string; however, the entropic magnetic force in this case will be zero due to the fact that the gradient of entropy along the string is absent. 

It is instructive to compare the entropic magnetic force with the bulk forces acting on the body embedded into magnetic field, and arising from the second term of Equation (2). Consider a diamagnetic body embedded into magnetic field $\vec{B}$; the bulk repulsion force $F_{magn}^{bulk}$ acting on the body is given in Equation (13): (13)$$F_{magn}^{bulk}=\int \int \int_{V}^{}\Delta \chi \frac{\vec{B}\cdot \nabla \vec{B}}{\mu_{0}}dV≅\Delta \chi \frac{\vec{B}\cdot \nabla \vec{B}}{\mu_{0}}V$$
where *V* is the volume of the body, and Δχ is the contrast in magnetic susceptibilities [14]. Let us compare the bulk and entropic magnetic force and define the dimensionless constant Ψ according to Equation (14):(14)$$\Psi =\frac{F_{magn}^{en}}{F_{magn}^{bulk}}$$ Consider diamagnetic body with a volume of *V* embedded into the magnetic field produced by the infinite direct current *I*, as shown in Figure 1. Taking into account: $\vec{B}≅\mu_{0}\vec{H;}H=\frac{I}{2\pi r}$ and simple calculations yield:(15)$$\Psi =\frac{N}{V}\frac{\mu_{0}\mu^{2}}{\Delta \chi k_{B}T}=\frac{\mu_{0}n\mu^{2}}{\Delta \chi k_{B}T}$$
where $n=\frac{N}{V}$ is the volume concentration of elementary magnets. Quite remarkably, the dimensionless number Ψ is independent on the spatial location of the diamagnetic body, and it scales with temperature as $\Psi =\frac{const}{T}$ (consider that magnetic susceptibility is practically independent on the temperature for diamagnetic materials); thus, the entropic repulsion is expected to prevail on the bulk magnetic force under the low temperatures.

### 2.3. Entropic Magnetic Forces in the Field Produced by the Magnetic Dipole 

It is also instructive to calculate the entropic magnetic force acting on the string of elementary magnets *μ* embedded into the field produced by the magnetic moment (current loop) $\vec{M}$ located, as shown in Figure 3. In this case, $H=\frac{M}{4\pi r^{3}}$; substitution into Equation (5) yields:(16)$$S_{i}=S_{0i}-\frac{\tilde{N}\mu_{0}^{2}\mu^{2}M^{2}dr_{i}}{32\pi^{2}k_{B}T^{2}r_{i}^{6}}$$ Integration of Equation (5) (*s* demonstrated above) and calculation of the entropic force yield for r≫L: (17)$$S=S_{0}-\frac{N\mu_{0}^{2}\mu^{2}M^{2}}{32\pi^{2}k_{B}T^{2}r^{6}}$$
(18)$$F_{magn}^{en}=\frac{3N\mu_{0}^{2}\mu^{2}M^{2}}{16\pi^{2}k_{B}Tr^{7}}$$ Again, the magnetic entropic force is the repulsion force, and it scales as $F_{magn}^{en}=\frac{const}{T}$. It will be also instructive to calculate the dimensionless number Ψ describing interrelation of the bulk and entropic magnetic forces, acting on the diamagnetic string and introduced in the previous section. For the string of elementary magnets embedded in the magnetic field produced by the magnetic dipole, we calculate: (19)$$\Psi =\frac{9}{\pi^{2}}\frac{\mu_{0}n\mu^{2}}{\Delta \chi k_{B}T}≅\frac{\mu_{0}n\mu^{2}}{\Delta \chi k_{B}T}$$
which coincides within the numerical factor close to unity with that given by Equation (15). Again, the dimensionless number Ψ supplied by Equation (19) is independent on the spatial location of the diamagnetic body, and it scales with temperature as $\Psi =\frac{const}{T}$; thus, the entropic magnetic repulsion will prevail on the bulk magnetic force under the low temperatures.

The force supplied by Equation (18) corresponds to the entropic potential energy of interaction between the string of elementary magnets *μ* and magnetic moment (current loop) $\vec{M}\;:$(20)$$U_{magn}^{en}\left(r\right)=\frac{N\mu_{0}^{2}\mu^{2}M^{2}}{32\pi^{2}k_{B}Tr^{6}}+const$$

## 3. Conclusions 

We report the entropic force emerging in the linear system of *N* elementary non-interacting magnets (spins $\vec{\mu }$) exerted to the external magnetic field $\vec{H}$, which tends to order the spins, and consequently to diminish the entropy of entire system. Thermal equilibrium with the thermal bath *T* is adopted. We considered the string of elementary non-interacting magnets with the length of *L*, exposed to the permanent magnetic field generated by straight infinite current *I*. The calculation of the magnetic entropic force, arising from the entropy gradient, yielded the expression: $F_{magn}^{en}=\frac{N\mu_{0}^{2}\mu^{2}I^{2}}{4\pi^{2}k_{B}Tr^{3}}$, where *r* is the distance between the current *I* and the edge of the string of elementary magnets (spins), which holds for the “long distance” approximation, namely when r≫L is adopted. The entropic magnetic force is the repulsion force. Somewhat surprisingly, it scales as $F_{magn}^{en}=\frac{const}{T}$; consider that the entropic forces usually grow with temperature. This prediction becomes understandable if we take into account that the temperature inspired chaos withstands the magnetic ordering of spins, and thus prevents their magnetic ordering imposed by the external magnetic field; hence, the increase in temperature is expected to diminish the magnetic entropic force. The magnetic entropic force scales as $F_{magn}^{en}\sim N$, which is also quite intuitively clear: the growth of the number of the spins strengthens the entire entropic magnetic effect. The effect of “entropic pressure” is predicted for the situation when the current is embedded into the continuous media, comprising elementary magnets/spins. The exact expression for the entropic magnetic forces, which is valid for the entire range of distances, is supplied. Entropy forces acting on the 1D string of elementary magnets exposed the magnetic field produced by the magnetic dipole are treated. 

We also studied interrelation between bulk and entropy magnetic forces. The dimensionless number $\Psi =\frac{F_{magn}^{en}}{F_{magn}^{bulk}}=\frac{\mu_{0}n\mu^{2}}{\Delta \chi k_{B}T}$ describing this interrelation was introduced. The value of Ψ is remarkably independent on the spatial location of the diamagnetic body, and it scales with temperature as $\Psi =\frac{\mathrm{const}}{T}$; thus, the entropic magnetic repulsion is expected to prevail on the bulk magnetic force under the low temperatures. Entropy forces, acting on the 1D string of elementary magnets exposed the magnetic field produced by the magnetic dipole, are addressed.

## Figures and Tables

**Figure 1 entropy-24-00299-f001:**
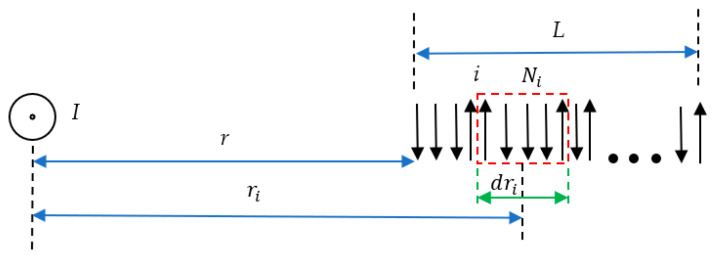
The linear system (string) of elementary magnets shown with arrows exerted to the permanent magnetic field generated by the infinite direct current *I* is depicted. Current *I* is perpendicular to the plane of the drawing. The length of the string is *L*; the distance between the current and the edge of the string is *r*.

**Figure 2 entropy-24-00299-f002:**
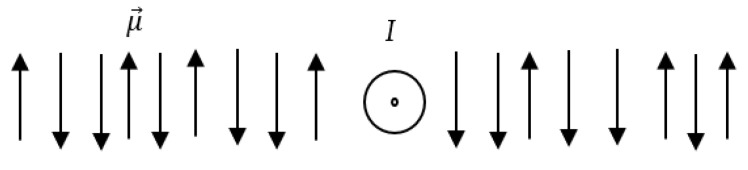
Direct current *I* embedded into the medium built of magnetic moments $\vec{\mu }$. The magnetic entropic force given by Equation (9) will repulse the magnetic moments, thus giving rise to the “entropic pressure phenomenon”.

**Figure 3 entropy-24-00299-f003:**
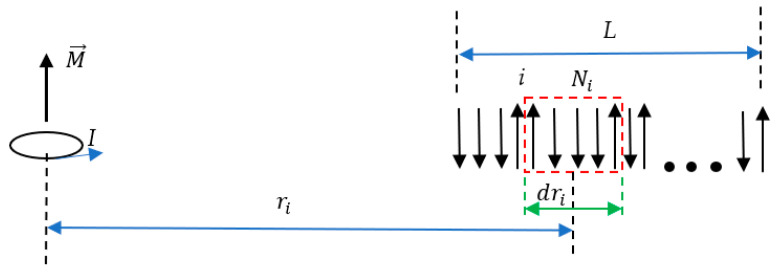
The linear system (string) of elementary magnets shown with arrows exerted to the permanent magnetic field generated by the magnetic dipole $\vec{M}$
*I* is depicted. The length of the string is *L*; the distance between the center of the magnetic dipole and the edge of the string is *r*.

## Data Availability

Not applicable.

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
