# Peer review of "Magnetic Entropic Forces Emerging in the System of Elementary Magnets Exposed to the Magnetic Field"

_entropy, 2022, doi:10.3390/e24020299_

Round 1

Reviewer 1 Report

Comments on 

Magnetic Entropic Forces Emerging in the System of Elementary Magnets Exposed to the Magnetic Field 3 by Edward Bormashenko:

are in the file attached!

Author Response

The author is thankful to the anonymous reviewer for extremely useful and fruitful remarks, which definitely improved the manuscript. The paper was revised and corrected according to the guidelines suggested by the respectable reviewer (see the attached detailed reply to reviewer, please). The paper was clarified and essentially expanded under the revision (see the revised version of the manuscript, please). 

The list of references was expanded and corrected. The paper was additionally spelled and edited. The Figures were improved.

The author is thankful for extremely fruitful and instructive reviewing of the manuscript.

Sincerely,

Professor Edward Bormashenko

16 February 2022.

Reviewer 2 Report

This manuscript on “Magnetic Entropic Forces Emerging in the System of Elementary Magnets Exposed to the Magnetic Field” provides a brief account of essentially a single result (Eq. 9) that has some merit in the entropic forces discussion, but currently is presented in a way that is inappropriate for publication. Proper introduction and discussion are lacking, while other parts are unnecessary repeated, as if to extend the text. I suggest resubmission of this manuscript as a Communication or Technical Note after major revisions.

Major comments:

Abstract: “The entropic force at large distance from the current is given by … is adopted” Leave formulas out of the abstract, especially as this is a limiting case only. Why not stress that an exact result has been found?

The introduction and references list [7-12] puts too much focus on entropic gravity. The latter can be mentioned as an analogue, but the introduction and discussion are as such incomplete. Most importantly, both sections should discuss this work with respect to the 'standard' view on magnetic repulsion as an electromagnetic force (similar to how entropic gravity is discussed in comparison with Newtonian / Einsteinian gravity).

Line 72: Explain the role and origin of S_0i

Lines 130-131: “Consider a string of elementary magnets embedded into the ideal solenoid” This can actually be done in several ways, affecting the discussion.

Sections 2 and 3: Stating that “the effect of “entropic pressure” is predicted” sounds quite exaggerated. A proper discussion on how it is here newly “understood” or “explained” would be more appropriate.

Minor comments:

Line 80: (consider Eq. 1 and N = const) is twice the same.

Line 96: Replace “whenever” by “wherever”

Lines 123-127: The text structure would be more logical if this part including Eq. 9 is put between equations 4 and 5, i.e., having the exact result first, and then providing a limiting case.

Author Response

The author is thankful to the anonymous reviewer for extremely useful , instructive and fruitful remarks, which definitely improved the manuscript. The paper was revised and corrected according to the guidelines suggested by the respectable reviewer (see the attached detailed reply to reviewer, please). The paper was clarified and essentially expanded under the revision (see the revised manuscript, please).

The list of references was expanded and corrected. The paper was additionally spelled and edited. The Figures were improved.

The author is thankful for extremely fruitful and instructive reviewing of the manuscript.

Sincerely,

Professor Edward Bormashenko

16 February 2022.

Round 2

Reviewer 1 Report

The quality of the paper has been greatly improved. I have no further objections and can recommend the paper for publishing.

Author Response

The author is thankful for a thorough and instructive reviewing of the manuscript.

The paper was additionally corrected and spelled.

Very sincerely,

Professor Edward Bormashenko

Reviewer 2 Report

The author has taken into account all reviewer comments and the manuscript has hence much improved, showing several results and their detailed discussion. A few minor shortcomings remain:

  • Writing can be improved; it would be helpful to have the manuscript edited by a native speaker. E.g. when used as an adjective, one always needs "entropic" (not "entropy")
  • I still have difficulties with the indication of "entropic pressure" being 'predicted' by the theory. This 'prediction' takes place 'post-factum' so is remains rather an "anticipation"
  • The new Section 2.4 is way too brief, raising doubts on a 'complex' subject that would require a publication on its own. I suggest to remove this subsection to keep the work focused and consistent.

Author Response

The author is thankful for an instructive reviewing of the paper.

We agreed with the respectable reviewer and excluded Section 2.4 from the final version of the manuscript.

Thank you very much for a fruitful reviewing of the paper,

Professor Edward Bormashenko

17 February 2022